# Gene expression and soluble protein level of PD-1 and its ligands (PD-L1 and PD-L2) in endometrial cancer

**Mohd Nazzary Mamat @ Yusof**📍, **Kah Teik Chew**📍*, **Nirmala Chandralega Kampan**, **Abdul Muzhill Hannaan Abdul Hafizz**📍, **Mohamad Nasir Shafiee**

Department of Obstetrics and Gynaecology, Faculty of Medicine, Universiti Kebangsaan Malaysia, Kuala Lumpur, Malaysia

* drchewkt@gmail.com

## Abstract

Checkpoint programmed death-1 (PD-1) and programmed cell death ligands (PD-Ls) are negative immunoregulatory molecules that assist tumour cells in evading the immune system. The interaction of PD-1 and PD-Ls inhibits T cells and tumour-infiltrating lymphocytes (TILs) while increasing the function of immunosuppressive regulatory T cells (Tregs). This leads to the evasion of the immune response by tumour cells. The roles of PD-1, PD-L1, and PD-L2 in endometrial cancer (EC) have not been fully elucidated. This study investigates the mRNA gene expression and soluble protein levels of these molecules in EC compared to controls, with detailed analysis of clinical profiles. The results showed that EC had significantly higher mRNA gene expression and soluble protein levels of PD-L1 and PD-L2, but not PD-1. Specifically, PD-1 mRNA gene expression was significantly higher in cases with less than 50% myometrial invasion. Additionally, the soluble protein level of PD-1 was substantially higher in patients under the age of 60. Higher gene expression of PD-L1 was observed only in advanced stages of EC. However, the soluble PD-L1 protein level was significantly elevated in type II EC, advanced stage, higher grade, lympho-vascular space invasion (LVSI), and in cases with myometrial invasion of 50% or more. PD-L2 mRNA gene expression and soluble protein levels significantly differed across all clinical profiles except for LVSI. These findings suggest that PD-1, PD-L1, and PD-L2 may serve as potential predictive biomarkers, which could be beneficial for the management of endometrial cancer patients through immunotherapy.

## Introduction

Endometrial cancer (EC), also known as uterine cancer, originates in the lining of the uterus (endometrium). While its primary demographic comprises postmenopausal women, occurrences among younger individuals are not uncommon. The incidence of EC has exhibited a consistent upward trend, posing a substantial global public health concern [1,2]. In 2020, 417,336 cases of EC were reported around the world and reported as the sixth most common cancer in women. Predominantly, EC manifests in women aged 65 to 75 years. Disparities in race, socioeconomic status, and geographical location significantly influence the frequency and mortality rates attributed to EC [1,3,4].

**Data availability statement:** All relevant data are within the paper.

**Funding:** This study received funding from the Fundamental Research Grant Scheme by the Ministry of Higher Education Malaysia, FRGS/1/2020/SKK0/UKM/03/2.

**Competing interests:** The authors have declared that no competing interests exist.

Although the Cancer Genome Atlas (TCGA) has significantly improved our understanding of EC's biological heterogeneity, precise molecular classification for surgical staging, adjuvant therapy, and surveillance strategies remains unclear. A comprehensive understanding of host factors such as microbiome composition and body mass index (BMI) is important for the effective management of EC. Additionally, delineation of the molecular and immunological determinants underlying responses to emerging therapies and the development of resistance is crucial for the design of next-generation studies. Moreover, considerations of therapeutic efficacy, survival outcomes, quality of life (QoL), and economic implications are pivotal facets that must be incorporated into comprehensive management strategies [1].

Large-scale genomic research has improved EC carcinogenesis insight and integrated molecular biomarkers into global risk classification systems. Innovative immunotherapeutic drugs targeting specific molecules in DNA Polymerase Epsilon, catalytic subunit (POLE) mutant genes, or tumours with microsatellite instability (MSI) are also being developed [5,6]. Staging assessments, which evaluate the primary tumour, lymph nodes, and distant metastases, are essential for controlling tumour progression. Surgical resection is the primary treatment for EC, and minimally invasive surgery (MIS) offers extra benefits. In some cases, EC is treated with pelvic and para-aortic lymphadenectomy. Within the surgical management of EC, sentinel lymph node biopsy (SLNB) is employed to map the primary lymphatic drainage of the uterus, enabling early detection of metastases and reducing mortality [5,6].

As cancer progresses, it often undergoes several changes. These mutations provide cancer cells with an evolutionary advantage by increasing genetic diversity, thereby accelerating their evolutionary fitness. However, this genetic diversity comes at a cost: the more distinct a cancer cell is from a normal cell, the more likely it is to be recognized as foreign by the immune system [7,8]. Previously, the immune system's role in combating cancer was not well understood because tumour cells disguise themselves by altering their appearance, making them harder for the immune system to recognize [8,9]. In recent decades, research has focused on blocking immunoreceptors, such as programmed cell death protein 1 (PD-1) and its ligands (PD-L1 and PD-L2), in various cancers. Immune checkpoints, which consist of molecules that regulate the immune response, have become a significant area of study [8,10–13].

Recent research has shown that initial treatment with anti-PD-1/PD-L1 therapy yields response rates ranging from 20% to 65% in tumours that are positive for PD-L1, particularly in EC [14–16]. In contrast, cancers lacking PD-L1 expression exhibit response rates varying from 0% to 17% across different types of malignancies [2,17]. PD-L1 expression inside the tumour microenvironment is widely acknowledged as a crucial molecule for identifying individuals more likely to benefit from immunotherapy treatment [2].

Research of immunomodulation in EC has emerged as an important area in recent years. In 2017, Pembrolizumab received approval from the FDA for the treatment of solid tumours characterized by either unresectable or metastasis to various anatomical sites, alongside demonstrating deficient mismatch repair (dMMR) [2]. Anti-PD-1 and anti-PD-L1 drugs, such as nivolumab, durvalumab, and dostarlimab, have been robustly subject to investigation within the realm of immune checkpoint inhibitors [18–21]. The finding shows that immunotherapy has potential significant benefit in certain EC patients, providing hope for those with advanced or recurrent conditions [2].

The latest findings from the RUBY and NRG-GY018 studies indicate that the concurrent administration of immune checkpoint inhibitors (ICIs), such as dostarlimab or pembrolizumab, with conventional chemotherapy yields a substantial enhancement in progression-free survival rates among patients diagnosed with advanced endometrial cancer [22–24]. The combination demonstrates notable efficacy in individuals diagnosed with dMMR tumours while also conferring advantages to those afflicted with mismatch repair-proficient (pMMR) illness.

The results above underscore the growing significance of immune ICIs as a primary treatment option for advanced and metastatic endometrial cancer, especially in individuals with high microsatellite instability (MSI-H) or dMMR [23]. Patient groups with a substantial tumour mutational burden (TMB) have had favourable responses to immunomodulators targeting immune regulatory pathways, such as PD-1/PD-L1. This finding has led to a significant transformation in treatment paradigms since ICIs have exhibited enhanced efficacy and potentially superior survival outcomes compared to conventional chemotherapy [23].

The management of endometrial cancer is undergoing a swift transformation. Immunotherapy, biomarker-guided decisions, and personalized approaches can significantly enhance the quality of life and patient outcomes. Hence, this study directed its focus towards EC patient sample, aiming to analyse the mRNA gene expression and soluble protein levels of PD-1, PD-L1, and PD-L2 using quantitative polymerase chain reaction (qPCR) and enzyme-linked immunosorbent assay (ELISA) methods. This exploration aimed to establish associations conducive to the advancement of personalized immunotherapy.

## Materials and methods

### Ethics approval and sample collection

This study received approval from the Universiti Kebangsaan Malaysia (UKM) human ethics committee with the approval code UKMPP1/111/8/JEP-2021-455, in accordance with the principles of the Declaration of Helsinki. Written informed consent was obtained from each patient who agreed to participate in this study. Comprehensive medical histories and clinical information were documented for all patients. A total of 34 patients were recruited for this prospective convenient case-control study between 1 September 2021, and 31 January 2023, comprising of patients diagnosed with EC (n = 23) and control group (n = 11) for both tissue and blood samples.

The cancer group was collected from women with newly diagnosed endometrial cancer via endometrium sampling histopathological examination (HPE) undergone total abdominal hysterectomy (TAH). The exclusion criteria for cancer group are as follows: concurrent cancer; under chemotherapy, radiotherapy, or hormonal therapy; not having major surgery, open biopsy, or significant trauma or injury within 28 days before sampling. As for the control group, samples were collected from benign cases with no significant uterine or ovarian pathology detected on imaging. Endometrium sampling taken via pipelle sampling will be histologically confirmed as normal endometrium before enrolment into the study. The exclusion criteria for control group are as follows: on hormonal contraceptives; receiving nonsteroidal anti-inflammatory drugs (NSAIDs), anti-inflammatory steroids, or immunosuppressant agents within 14 days before sampling; not having major surgery, open biopsy, or significant trauma or injury within 28 days before sampling.

### RNA extraction from endometrium tissue, cDNA conversion and qPCR analysis

Endometrium tissue biopsy samples were immersed in RNAprotect Tissue Reagent (Qiagen GmBH, Hilden, Germany, cat. no. 76104) and kept at −80 °C freezer until further analysis. Total RNA was extracted from frozen tissue with RNeasy Mini Kit (Qiagen GmBH, Hilden, Germany, cat. no. 74104) according to manufacturer instructions. For control samples, before RNA extraction, the pathologist evaluated the tissue samples by performing H&E staining on a tissue section to inspect for the presence and distribution of immune cells visually. The purity and quantity of RNA were detected using the DS-11 + Spectrophotometer (DeNovix Inc. Wilmington, DE, USA). Subsequently, cDNA was synthesized with QuantiTect Reverse Transcription Kit (Qiagen GmBH, Hilden, Germany, cat. no. 205311) following the manufacturer's protocol.

Real-time quantitative polymerase chain reaction (qPCR) was performed on CFX96 Real-Time PCR (Bio-rad) with the cycling conditions consisting of 2 min PCR initial activation step at 95 ℃, 35 cycles at 95 ℃ for 5 seconds for the denaturation step and 10 seconds at 60 ℃ for combined annealing and extension step. GAPDH and ACTB were used as reference genes. Each 20 µl PCR reaction contained one µl of cDNA, 10 µl of the 2x QuantiNova SYBR Green PCR Master Mix (Qiagen GmBH, Hilden, Germany, cat. no. 208054), and 10 µM of forward and reverse primers. The primer sequences are as follows: PD-1 forward/reverse (new design): TTCCAGTGGCGAGAGAAGAC/GCCAA-GAGC AGTGTCCATCC, PD-L1 forward/reverse [25]: TATGGTGGTGCCGACTACAA/TGCTTG TCCAGATGACTTCG, PD-L2 forward/reverse [25]: GTACATAATAGAGCATGGCAGCA/CCACCTTTTGCAAACTGGCTGT, GAPDH forward/reverse [26]: TCAAGGCTGAGAACGG-GAAG/TCGCCCCACTTGATTTTGGA, ACTB forward/reverse [27]: AGGTGACAGCAGTCG-GTTGGA/CCTTAGAGAGAAGTGGGGTGG. Each experiment was duplicated to minimize errors, and the fold change equation was used to calculate the relative mRNA gene expression.

## Serum isolation, storage, and soluble protein analysis by ELISA

The venous blood samples collected from each group were centrifuged at acceleration-6 and break-3, 3000 rpm for 10 minutes at 25°C. Then, the serum was stored in a -80°C freezer until further analysis. The ELISA protein detection and quantification for soluble PD-1, PD-L1 and PD-L2 proteins was performed using serum samples from both EC group and the control group. The ELISA assayed following the manufacturer's protocol by a commercially available Human PD-1/PDCD1 (Programmed Cell Death Protein 1) (Finetest, cat. no. EH0252), Human PDCD1LG1 (Programmed Cell Death Protein 1 Ligand 1) (Finetest, cat. no. EH3528), and Human PDCD1LG2 (Programmed Cell Death Protein 1 Ligand 2) (Finetest, cat. no. EH4187), implementing sandwich enzyme immunoassay.

Briefly, 96-well plates were incubated for 90 minutes at 37 ℃ with 100 µl each of standard, blank, and sample with dilution 1:2 in duplicate. After incubation, microwell plates were aspirated and washed two times with the wash buffer. Then, 100 µl of working solution

Biotin-conjugate antibodies were added to all wells and incubated at 37 ℃ for 60 minutes. After aspirating and washing three times, 100 µl of working solution, HRP-Streptavidin Conjugate, was added and incubated for 30 minutes at 37 ℃. Then, the microplate was aspirated and washed five times. The colour development took place by adding 90 µl of the TMB substrate solution to all the wells and incubation at 37 ℃ in the dark for 20 min. Then, 50 µl of the stop solution was added to all wells, and the colour turned yellow can be seen. The absorbance of both the standards and samples was measured at 450 nm using a microplate reader.

## Statistical analysis

Statistical analyses were conducted using GraphPad Prism 9.1.0 (GraphPad Software Inc.). The data was checked for normal distribution using the Shapiro normality test. The significance of differences between the two groups was analysed using the Mann-Whitney U test, and multiple group comparisons were analysed using ANOVA followed by post-hoc Tukey's test for normally distributed data or Kruskal-Wallis test with Dunn's post-hoc test for not normally distributed data. The P values < 0.05 proved to be statistically significant.

## Results

### Clinical and pathological characteristics

The mean age of the 23 EC patients was $63 \pm 15$ years old. The age distribution within this group was as follows: fifteen were age old than 60 years, and eight were 60 years old and younger. Control samples collected are from women aged between 30 and 60 years old, with half of them

being menopausal. From a histological perspective, type I endometrial cancer (endometrioid) was present in 19 patients, while type II endometrial cancer (non-endometrioid) was identified in 3 patients. In terms of cancer grade, 15 patients were classified as low grade (grade I and II) and 4 cases as high grade (grade III). Regarding cancer staging, 20 cases were in early cancer stage, while the remaining 3 were in advanced cancer stages. Lympho-vascular space invasion (LVSI) was reported in 7 cases, whereas no invasion was observed in the remaining 16 cases. Additionally, myometrium invasion exceeding 50% was noted in 13 cases, while the remaining 10 cases had myometrium invasion below 50%.

## mRNA gene expression of PD-1, PD-L1, and PD-L2

PD-L1 gene expression was significantly increased in EC group (2.615-fold change) compared to the control group (1.188-fold change) ($p = 0.042$). PD-L2 gene expression also significantly increased in EC group (3.724-fold change) compared to the control group (1.211-fold change) ($p = 0.021$). However, there was no significant difference between EC and control group in terms of PD-1 gene expression ($p = 0.699$). Fig 1 illustrates the gene expression findings.

## Clinicopathological factors and mRNA gene expression of PD-1, PD-L1, and PD-L2

PD-1 and PD-L1 gene expression did not significantly differ between age groups (≤60 years old, >60 years old) among EC group, or compared to the control group ($p > 0.050$). In contrast, the PD-L2 mRNA gene expression showed a significant difference between the two age groups and the control group ($p = 0.035$). The post-hoc analysis of PD-L2 gene expression showed patients aged 60 and below had significantly higher expression than control but not significantly different with age older than 60, as shown in Table 1.

The histological type of EC did not show significant difference in mRNA gene expression of PD-1 ($p > 0.050$) and PD-L1 ($p > 0.050$) between type I, type II, and control groups. Significant differences were observed in PD-L2 gene expression compared to the two histological types and the control group ($p = 0.024$). Specifically, PD-L2 expression was significantly higher in type I EC compared to the control group. However, there was no significant difference in PD-L2 expression between type II EC and the control group, as indicated by post-hoc analysis (Table 1).

In terms of EC grade, only PD-L2 mRNA expression demonstrated significant difference between low grade, high grade, and control groups ($p = 0.030$). Specifically, PD-L2 expression

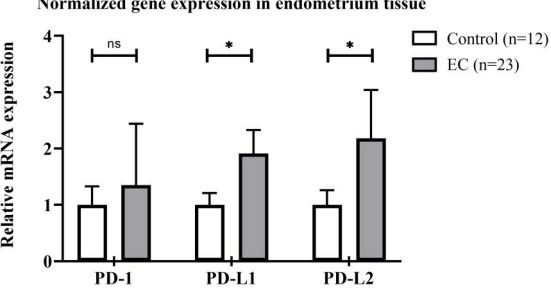

**Fig 1. Comparison of PD-1, PD-L1, and PD-L2 mRNA gene expression level (fold change) in endometrium tissue from control group and EC patients.** The levels of mRNA expression have been normalized with housekeeping gene (GAPDH and ACTB). Mann–Whitney U test was used to compare the expression. * indicates $p < 0.050$. ns indicates not significantly different.

**Table 1. Clinicopathological factors and mRNA gene expression of PD-1, PD-L1, and PD-L1 (fold change).**

| Variables | No. of sample (N) | mRNA gene expression, fold change | | | | | |
|---|---|---|---|---|---|---|---|
| | | PD-1, Mean (SD) | p-value | PD-L1, mean (SD) | p-value | PD-L2, mean (SD) | p-value |
| Age group | | | | | | | |
| Control | 11 | 1.433 (1.066) | 0.658 | 1.188 (0.719) | 0.119 | 1.212 (0.897)[a] | 0.035* |
| ≤60 | 8 | 5.389 (5.185) | | 2.818 (2.517) | | 4.839 (4.696)[b] | |
| >60 | 15 | 3.332 (5.456) | | 2.508 (1.872) | | 3.129 (3.962)[ab] | |
| Histological type | | | | | | | |
| Control | 11 | 1.433 (1.066) | 0.292 | 1.188 (0.719) | 0.067 | 1.212 (0.897)[a] | 0.024* |
| Type I | 19 | 4.767 (5.623) | | 2.255 (1.378) | | 4.219 (4.463)[b] | |
| Type II | 3 | 0.713 (0.790) | | 3.087 (3.669) | | 1.383 (1.399)[ab] | |
| Grade | | | | | | | |
| Control | 11 | 1.433 (1.066) | 0.535 | 1.188 (0.719) | 0.084 | 1.212 (0.897)[a] | 0.030* |
| Low Grade | 15 | 4.239 (5.052) | | 2.373 (1.375) | | 4.281 (4.741)[b] | |
| High Grade | 4 | 6.745 (8.000) | | 1.810 (1.494) | | 3.988 (3.810)[ab] | |
| Stage | | | | | | | |
| Control | 11 | 1.433 (1.066) | 0.837 | 1.188 (0.719)[a] | 0.042* | 1.212 (0.897)[a] | 0.020* |
| Early | 20 | 4.105 (5.585) | | 2.362 (1.903)[a] | | 2.362 (1.903)[ab] | |
| Advance | 3 | 3.663 (4.116) | | 4.307 (2.757)[a] | | 4.307 (2.757)[b] | |
| Lympho-vascular space invasion | | | | | | | |
| Control | 11 | 1.433 (1.066) | 0.867 | 1.188 (0.719) | 0.084 | 1.212 (0.897) | 0.057 |
| Yes | 7 | 3.740 (4.729) | | 2.377 (2.457) | | 2.810 (3.138) | |
| No | 16 | 4.182 (5.727) | | 2.720 (1.951) | | 4.123 (4.632) | |
| Myometrium invasion | | | | | | | |
| Control | 11 | 1.433 (1.066)[ab] | 0.015* | 1.188 (0.719) | 0.092 | 1.212 (0.897)[a] | 0.034* |
| ≥50% | 13 | 2.099 (3.806)[a] | | 2.235 (1.786) | | 2.420 (2.445)[ab] | |
| <50% | 10 | 6.580 (6.144)[b] | | 3.110 (2.388) | | 5.418 (5.445)[b] | |

One way ANOVA test or Kruskal Walis were used to determine the difference between the groups. * indicates p < 0.050. The label a and b indicate statistically different groups by post-hoc analysis, Turkey post-hoc for normally distributed data and Dunn post-hoc for not normally distributed data.

was significantly higher in the low-grade EC group compared to the control group. However, there was no significant difference in PD-L2 expression between the high-grade EC group and the control group. (Table 1).

The stages of EC exhibited significant differences in PD-L1 ($p = 0.042$), and PD-L2 ($p = 0.020$) mRNA gene expression compared to the control group. However, there was no significant difference observed for PD-1 mRNA gene expression ($p > 0.050$) between the EC stages and the control group. Further post-hoc analysis found that PD-L2 gene expression was significantly higher in the advanced stage compared to the control but not in early stages (Table 1).

No significant difference was found in mRNA gene expression of PD-1, PD-L1, and PD-L2 with respect to LVSI ($p > 0.050$). In the case of myometrium invasion, significant results were found for the mRNA gene expression of PD-1 ($p = 0.015$) and PD-L2 ($p = 0.034$). The post-hoc analysis revealed that cases with myometrium invasion of 50% or more had significantly higher PD-1 expression compared to cases with myometrium invasion of less than 50%. However, neither of these groups showed a significant difference compared to the control group. For the PD-L2 expression, the post-hoc analysis reported that the cases of myometrium invasion of more than 50% were significantly higher than the control group but not significantly differ from the myometrium less than 50% invasion (Table 1).

## Soluble protein level of PD-1, PD-L1, and PD-L2 in serum

Soluble protein level of PD-L1 in serum was significantly increased in EC (154.1 ± 105.1pg/mL) compared to control group (102.8 ± 23.21pg/mL) ($p = 0.025$). Similarly, the soluble protein level of PD-L2 was also significantly increased in EC (1331.0 ± 536.5pg/mL) compared to control (171.8 ± 149.7pg/mL) ($p < 0.001$). However, there was no significant difference between EC and control group for soluble protein level of PD-1 ($p = 0.572$). Fig 2 illustrates the soluble protein level findings.

## Clinicopathological factors and soluble protein level of PD-1, PD-L1, and PD-L2

Soluble protein level of PD-1 exhibited significant differences among EC patients aged less than or equal to 60 years old, EC patients aged more than 60 years old, and the control group ($p = 0.009$). Similarly, significant differences were observed among these age groups with soluble protein level of PD-L2 ($p < 0.001$). In contrast, there was no significant difference observed in the soluble protein level of PD-L1 between the two age groups and the control group ($p > 0.050$). The post-hoc analysis of soluble protein levels of PD-1 revealed that patients aged 60 and below had significantly higher levels than those older than 60. However, both groups were not significantly different from the control group (Table 2).

The histological type of EC did not show a significant difference between type I, type II, and control for soluble protein level of PD-1 ($p > 0.050$). However, significant differences were observed for soluble protein level for PD-L1 ($p = 0.004$) and PD-L2 ($p < 0.001$) compared to these two histological types and the control group. The soluble protein level of PD-L1 was significantly higher in type II EC compared to the control group, but not significantly different compared to type I EC, as determined by post-hoc analysis. Regarding soluble protein level of PD-L2, the level was significantly higher in both histological types compared to the control group. However, post-hoc analysis did not reveal significant differences between type I and type II EC (Table 2).

Soluble proteins level of PD-L1 and PD-L2 exhibited a significant difference between low grade, high grade, and control groups ($p = 0.040$; $p < 0.001$, respectively). Specifically, the soluble protein level of PD-L1 was significantly higher in the high-grade EC group compared to

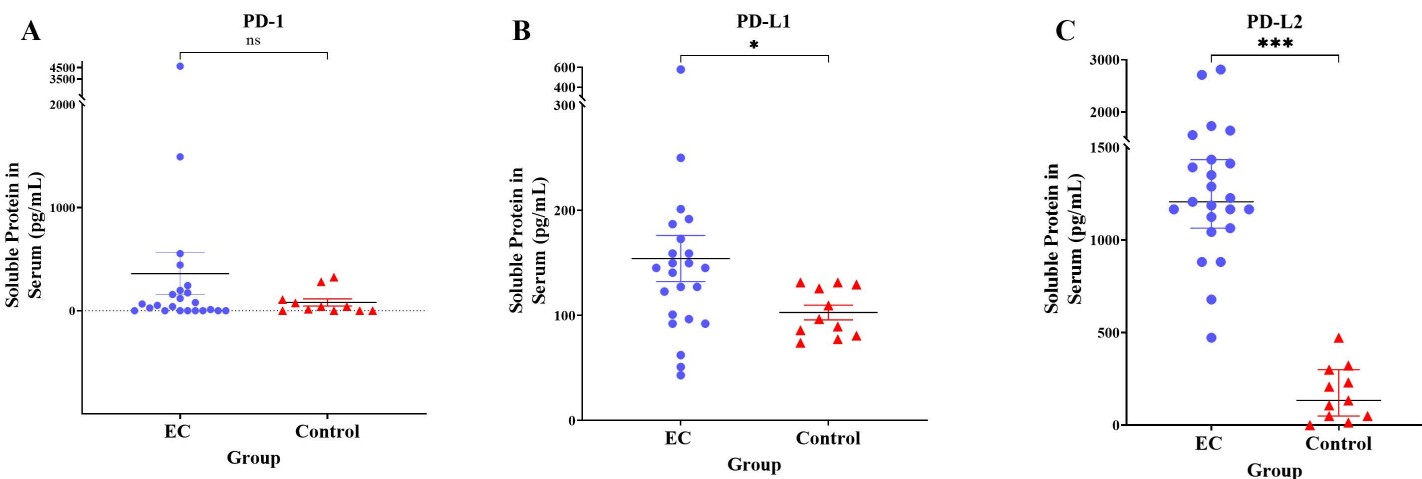

**Fig 2. Comparison of PD-1 (A), PD-L1 (B), and PD-L2 (C) soluble protein level (pg/mL) in serum of control and EC patients.** Mann–Whitney U test was used to compare the expression; * indicates $p < 0.050$; *** indicates $p < 0.001$; ns indicates not significantly different.

the control group. However, post-hoc analysis did not reveal a significant difference between the high-grade EC and low-grade EC groups. On the other hand, the soluble protein level of PD-L2 was significantly higher in both the low-grade EC and high-grade EC groups compared to the control group. However, post-hoc analysis did not reveal significant differences between the low-grade EC and high-grade EC groups (Table 2).

The soluble protein levels of PD-L1 and PD-L2 were significantly different across EC stages compared to the control group ($p = 0.027$; $p < 0.001$, respectively). Specifically, PD-L1 levels were higher in the advanced stage compared to the control group, but no significant difference was observed between the advanced and early stages. Soluble protein level of PD-L2 were elevated in both low-grade and high-grade EC groups compared to the control group, with no significant difference observed between different stages based on post-hoc analysis (Table 2).

Additionally, in the context of LVSI, PD-L1 and PD-L2 soluble protein levels were significantly different between patients with LVSI, without LVSI, and the control group ($p = 0.029$ and $p < 0.001$, respectively). PD-L1 soluble protein levels were higher among patients with LVSI compared to the control group, but no significant difference was observed between patients with LVSI and without LVSI. PD-L2 levels were elevated in both LVSI and non-LVSI

**Table 2. Clinicopathological factors and PD-1, PD-L1, and PD-L2 soluble protein level (pg/mL).**

| Variables | No. of sample (N) | Soluble protein level, pg/mL | | | | | |
|---|---|---|---|---|---|---|---|
| | | PD-1, mean (SD) | p-value | PD-L1, mean (SD) | p-value | PD-L2, mean (SD) | p-value |
| Age group | | | | | | | |
| Control | 11 | 80.14 (115.80)[ab] | 0.009** | 102.80 (23.21) | 0.050 | 171.80 (149.70)[a] | <0.001*** |
| ≤60 | 8 | 916.90 (1568.00)[a] | | 169.70 (171.20) | | 1382.00 (650.10)[b] | |
| ˃60 | 15 | 61.99 (116.70)[b] | | 145.80 (49.74) | | 1304.00 (488.60)[b] | |
| Histological type | | | | | | | |
| Control | 11 | 80.14 (115.80) | 0.788 | 102.80 (23.21)[a] | 0.004** | 171.80 (149.70)[a] | <0.001*** |
| Type I | 19 | 394.10 (1078.00) | | 149.40 (110.90)[ab] | | 1330.00 (580.10)[b] | |
| Type II | 3 | 74.40 (39.98) | | 214.20 (31.26)[b] | | 1314.00 (367.50)[b] | |
| Grade | | | | | | | |
| Control | 11 | 80.14 (115.80) | 0.359 | 102.80 (23.21)[a] | 0.040* | 171.80 (149.70)[a] | <0.001*** |
| Low Grade | 15 | 488.70 (1203.00) | | 148.40 (125.10)[ab] | | 1235.00 (526.60)[b] | |
| High Grade | 4 | 39.26 (78.52) | | 153.40 (25.85)[b] | | 1687.00 (713.90)[b] | |
| Stage | | | | | | | |
| Control | 11 | 80.14 (115.80) | 0.703 | 102.80 (23.21)[a] | 0.027* | 171.80 (149.70)[a] | <0.001*** |
| Early | 20 | 398.10 (1051.00) | | 152.20 (112.70)[ab] | | 1310.00 (565.10)[b] | |
| Advance | 3 | 100.60 (66.96) | | 166.80 (22.02)[b] | | 1473.00 (313.10)[b] | |
| Lympho-vascular space invasion | | | | | | | |
| Control | 11 | 80.14 (115.80) | 0.676 | 102.80 (23.21)[a] | 0.029* | 171.80 (149.70)[a] | <0.001*** |
| Yes | 7 | 108.40 (89.49) | | 156.20 (37.76)[b] | | 1336.00 (247.70)[b] | |
| No | 16 | 469.10 (1170.00) | | 153.20 (125.0)[ab] | | 1329.00 (630.60)[b] | |
| Myometrium invasion | | | | | | | |
| Control | 11 | 80.14 (115.8) | 0.772 | 102.80 (23.21)[a] | 0.016* | 171.80 (149.70)[a] | <0.001*** |
| ≥50% | 13 | 421.2 (1262) | | 182.60 (129.00)[b] | | 1465.00 (420.50)[b] | |
| <50% | 10 | 279.0 (470.6) | | 117.10 (45.84)[ab] | | 1156.00 (638.60)[b] | |

One way ANOVA test or Kruskal Walis were used to determine the difference between the groups. * indicates $p < 0.050$; ** indicates $p < 0.010$; *** indicates $p < 0.001$. The label a and b indicate statistically different groups by post-hoc analysis, Turkey post-hoc for normally distributed data and Dunn post-hoc for not normally distributed data.

patients compared to the control group, with no significant difference between different LVSI groups based on post-hoc analysis (Table 2).

In the case of myometrium invasion, significant results were demonstrated for the soluble protein levels of PD-L1 ($p = 0.016$) and PD-L2 ($p < 0.001$), but no significant difference was observed for PD-1 level ($p > 0.050$). Post-hoc analysis revealed that patients with myometrium invasion of 50% or more had significantly higher soluble protein levels of PD-L1 compared to the control group, but no significant difference was observed between patients with myometrium invasion of less than 50% and the control group. For the PD-L2 protein level, the post-hoc analysis reported that the patients of myometrium invasion of more than 50% were significantly higher than the control group but did not significantly differ from the myometrium less than 50% invasion (Table 2).

## Discussion

Data from this study provided a comprehensive analysis of mRNA gene expression and soluble protein levels of PD-1, PD-L1, and PD-L2 in EC patients compared to the control groups, along with sub-analysis with various clinicopathological parameters. The significant increase in PD-L1 and PD-L2 gene expression in EC patients compared to controls suggests their potential role in EC pathogenesis via immune evasion mechanisms [2,28,29]. Consistent with mRNA expression findings, soluble protein levels of PD-L1 and PD-L2 were significantly increased in EC patients compared to controls. This further supports the role of PD-L1 and PD-L2 in EC pathogenesis and suggests their potential as diagnostic or prognostic biomarkers [30,31].

While no significant differences in PD-1 and PD-L1 gene expression were observed across different age groups, PD-L2 mRNA expression showed a significant difference. Specifically, patients aged 60 and below exhibited higher PD-L2 expression compared to the control groups. Soluble protein levels of PD-1 and PD-L2 showed significant differences across age groups, with younger patients exhibiting higher levels compared to the control groups. These outcomes were supported by previous research on other cancers [30,32–34]. However, there is limited study of endometrial cancer regarding mRNA and soluble protein levels of PD-1 and its ligands, with no associations found from immunohistochemical (IHC) analysis of tumour tissues [11,12]. This age-related difference in gene expression and protein levels may reflect variations in immune response or tumours-host interactions, suggesting a potential stratification for personalized immunotherapy. Patients can be stratified based on their age-related expression patterns of PD-1 and its ligands. In particularly, younger patients with higher expression of PD-1 and its ligands may benefit from additional immunotherapy approaches [35]. This personalized approach enhances treatment efficacy and improve outcomes in this specific patient population.

PD-L2 gene expression exhibited significant discrepancies across histological types and cancer grades compared to the control group. Similarly, soluble PD-L2 protein levels, along with soluble PD-L1 protein levels, showed significant disparities between histological types and cancer grades. These findings provide additional support to the previously reported histological and grade-specific implications of PD-L1 (particularly in type II and high-grade tumours) and PD-L2 in the progression of EC, as highlighted in previous studies [8,11–13]. By understanding the intricate relationship between immune checkpoint molecules like PD-L1 and PD-L2 and the histological subtypes and grades of the disease, clinicians can tailor treatment regimens to target specific molecular pathways associated with tumour progression and immune evasion. Additionally, these findings may use as prognostic assessments, helping clinicians predict patient outcomes more accurately and identify individuals who may benefit

most from certain treatment, especially immunotherapy. Ultimately, this has the potential to improve treatment outcomes and enhance the overall survival rate of EC.

The analysis revealed significant differences in PD-L1 and PD-L2 gene expression levels across various stages of EC compared to controls. Interestingly, PD-L2 expression was found to be elevated in advanced-stage EC compared to controls. This observation suggests a potential correlation between PD-L2 expression and disease progression, highlighting its potential utility as a biomarker for advanced-stage EC. Moreover, soluble protein levels of PD-L1 and PD-L2 also showed significant differences across EC stages compared to controls, which aligns with previous findings for PD-L1 but not for PD-L2 [11,13]. This discrepancy suggests a potential association between soluble PD-L1 and PD-L2 levels and disease stage, with implications for disease monitoring and prognosis. These findings highlight the importance of investigating both gene expression and soluble protein levels of immune checkpoint molecules like PD-L1 and PD-L2 in understanding the dynamics of EC progression and its potential clinical implications for disease monitoring and prognosis.

The gene expression of PD-1 and PD-L2 exhibited significant differences based on myometrium invasion status compared to normal endometrium, whereas no significant difference was found for LVSI. However, soluble protein levels of PD-L1 and PD-L2 showed significant differences based on both myometrium invasion and LVSI status. It's noteworthy that these outcomes were not found in previous studies, which primarily utilized IHC methods [11–13]. In contrast, our study employed qPCR and ELISA techniques, which might offer greater sensitivity and specificity in detecting molecular changes associated with myometrium invasion and LVSI in EC. This underscores the importance of utilizing multiple analytical methods to comprehensively explore the role of immune checkpoint molecules like PD-1, PD-L1, and PD-L2 in EC progression and their potential as biomarkers for disease prognosis and management.

The present study provides valuable insights into how gene expression and soluble protein levels can serve as preliminary indicators, offering a comparative analysis to the gold-standard diagnostic tool of IHC for EC. Genomic rearrangements have been identified as the underlying mechanisms responsible for causing amplifications of the PD-1 gene and its ligands. These amplifications typically do not impact the open reading frame, resulting in a significant increase in expression. However, it's worth noting that there isn't always a consistent relationship between PD-L1 expression and gene copy number in specific malignancies. This suggests that certain type of EC may lack the necessary trans-activators to drive PD-L1 expression [36]. This highlight the complex regulatory mechanisms involved in immune checkpoint molecule expression and their implications for tumour biology and therapeutic targeting in EC [36].

Research has indicated that PD-1, PD-L1, and PD-L2 serum soluble protein levels serve as distinct prognostic variables in various solid tumours, including EC, susceptible to immunotherapy targeting the PD-1 axis. However, the regulation, origin, and function of EC's soluble PD-1, PD-L1, and PD-L2 remain subjects of debate [37–39]. Limited studies have examined serum levels of soluble PD-1 and its ligands in EC. Initially reported in autoimmune diseases, soluble protein of PD-1, PD-L1, and PD-L2 were believed to be generated by immune cells in response to proinflammatory cytokines. Interestingly, no cancer type has yet had these soluble proteins evaluated simultaneously with gene expression in current research advances. We believe that our current study is the first to comprehensively investigate the gene expression and soluble protein levels of PD-1, PD-L1, and PD-L2 simultaneously and correlate them with clinicopathological aspects in EC. This approach allows for a more holistic assessment of the immune microenvironment within the tumour and its implications for disease progression and treatment response. A recent study demonstrated a strong positive correlation between soluble PD-1 and PD-L1 levels in patients with advanced prostate cancer, with similar outcomes

observed in autoimmune disease. This finding suggests that soluble PD-1 and PD-L1 may originate from the same source and be regulated similarly in cancer patients [39,40].

The knowledge and management of endometrial cancer have been significantly transformed by recent advancements in molecular classification, surpassing the conventional histological classifications of types I and II. The characterisation of specific molecular subtypes, including POLE-ultramutated, MSI-H, copy-number low, and copy-number high (serous-like), has facilitated the development of more personalised treatment strategies [41,42]. These subtypes exhibit variations in genetic modifications and immunological profiles, which encompass differing levels of PD-L1 expression. The MSI-H and POLE-ultramutated tumours frequently have elevated levels of PD-L1 expression, resulting in enhanced responsiveness to ICIs. Conversely, copy-number low and high subtypes typically display low PD-L1 expression, making them less amenable to ICIs and requiring other therapeutic approaches [41,42].

Differential expression of PD-L1 is observed in multiple classifier endometrial cancer cases, characterised by the presence of two or more mutational patterns, including POLE mutations, dMMR/MSI-H, and p53 abnormalities [42]. Research findings show a notable increase in PD-L1 expression in these tumours, particularly in cases where dMMR/MSI-H status is present, associated with a more favourable ICI response. Given the intricate nature of the molecular landscape, PD-L1 may emerge as a significant biomarker in treatment outcome prediction [42].

Furthermore, a well-established association exists between dMMR, TMB, and PD-L1 expression, whereby tumours exhibiting dMMR frequently have elevated TMB levels, resulting in increased PD-L1 expression [23]. From a clinical perspective, evaluating MMR status in isolation can potentially identify patients who are favourable candidates for ICI therapy, particularly in cases when TMB testing is limited [23]. Integrating these indicators wherever feasible may achieve a more thorough evaluation, potentially enhancing patient selection and treatment results.

Emerging therapeutic approaches, such as selinexor, have demonstrated potential in treating wild type p53 (p53 wt) tumours. Selinexor, a pharmacological agent that selectively inhibits nuclear export, can reinstate the functionality of p53 by ionising it within the nucleus, enhancing its tumour suppressor activity [43,44]. This phenomenon has particular significance in tumours with p53 wild-type variants since they may exhibit favourable responses to selinexor irrespective of their PD-L1 status. In the context of PD-L1 negative tumours, selinexor presents a potentially advantageous monotherapy alternative. Conversely, in PD-L1 positive tumours, combining selinexor with ICIs can improve treatment results by explicitly targeting immune evasion mechanisms and reactivating pathways associated with tumour suppression [23,43,44]. Incorporating molecular indicators such as p53 status and PD-L1 expression is paramount in informing therapeutic approaches for endometrial cancer.

The current study has several limitations that should be acknowledged. Firstly, the sample size is relatively small, which could limit the generalizability of the findings. Larger sample sizes are typically preferred to ensure the robustness and representativeness of the results. Secondly, it is difficult to get a sample older than 60 years old for the control group, so further comparison analysis regarding the age group cannot be done. Thirdly, using different methods (qPCR and ELISA) to analyse the soluble proteins of PD-1 and its ligands may yield varying results. Differences in assay sensitivity, specificity, and detection limits can lead to discrepancies in protein level measurements, potentially impacting the interpretation and comparability of findings across studies. Future studies with more extensive cohorts are necessary to validate and expand upon these findings, providing a more comprehensive understanding of the relationships between gene expression, soluble protein levels, and clinicopathological characteristics in EC.

## Conclusion

mRNA gene expression and soluble protein levels of PD-L1 and PD-L2 were enhanced in EC, indicating the activation of immune escape mechanisms. The findings offer valuable insights into the involvement of PD-1, PD-L1, and PD-L2 in the progression of EC. These molecules have the potential to serve as biomarkers for the diagnosis, prognosis, and stratification of EC treatment. Furthermore, the observed connections with clinicopathological factors highlight the diversity of EC and the importance of individualized therapy strategies that focus on checkpoint pathways.

## Acknowledgments

Not applicable.

## Author contributions

**Conceptualization:** Mohd Nazzary Mamat @ Yusof, Kah Teik Chew.

**Data curation:** Mohd Nazzary Mamat @ Yusof, Abdul Muzhill Hannaan Abdul Hafizz.

**Formal analysis:** Mohd Nazzary Mamat @ Yusof, Abdul Muzhill Hannaan Abdul Hafizz.

**Funding acquisition:** Kah Teik Chew, Mohamad Nasir Shafiee.

**Investigation:** Mohd Nazzary Mamat @ Yusof.

**Methodology:** Mohd Nazzary Mamat @ Yusof, Nirmala Chandralega Kampan, Abdul Muzhill Hannaan Abdul Hafizz, Mohamad Nasir Shafiee.

**Project administration:** Kah Teik Chew, Mohamad Nasir Shafiee.

**Resources:** Mohd Nazzary Mamat @ Yusof, Kah Teik Chew, Nirmala Chandralega Kampan, Abdul Muzhill Hannaan Abdul Hafizz, Mohamad Nasir Shafiee.

**Software:** Mohd Nazzary Mamat @ Yusof, Abdul Muzhill Hannaan Abdul Hafizz.

**Supervision:** Kah Teik Chew, Nirmala Chandralega Kampan, Mohamad Nasir Shafiee.

**Validation:** Kah Teik Chew, Nirmala Chandralega Kampan, Mohamad Nasir Shafiee.

**Visualization:** Mohd Nazzary Mamat @ Yusof.

**Writing – original draft:** Mohd Nazzary Mamat @ Yusof, Abdul Muzhill Hannaan Abdul Hafizz.

**Writing – review & editing:** Mohd Nazzary Mamat @ Yusof, Kah Teik Chew, Nirmala Chandralega Kampan, Mohamad Nasir Shafiee.

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
