## [Decision Letter · Decision Letter 0]

29 Jul 2024

PONE-D-24-25776Gene Expression and Soluble Protein Level of PD-1 and Its Ligands (PD-L1 and PD-L2) in Endometrial CancerPLOS ONE

Dear Dr. Chew,

Thank you for submitting your manuscript to PLOS ONE. After careful consideration, we feel that it has merit but does not fully meet PLOS ONE’s publication criteria as it currently stands. Therefore, we invite you to submit a revised version of the manuscript that addresses the points raised during the review process.

We look forward to receiving your revised manuscript.

Kind regards,

Yasuhiro Miki

Academic Editor

PLOS ONE

Journal Requirements:

The Fundamental Research Grant Scheme by the Ministry of Higher Education Malaysia, FRGS/1/2020/SKK0/UKM/03/2.

Reviewers' comments:

Reviewer's Responses to Questions

**Comments to the Author**

1. Is the manuscript technically sound, and do the data support the conclusions?

Reviewer #1: Partly

Reviewer #2: Yes

2. Has the statistical analysis been performed appropriately and rigorously? 

Reviewer #1: No

Reviewer #2: Yes

3. Have the authors made all data underlying the findings in their manuscript fully available?

Reviewer #1: Yes

Reviewer #2: Yes

4. Is the manuscript presented in an intelligible fashion and written in standard English?

Reviewer #1: Yes

Reviewer #2: Yes

5. Review Comments to the Author

**Reviewer #1: ** Present article is so interesting and must be valuable for the patients suffering from uterine endometrial cancer. However, several concerns remain in this article, which suggest as follows:

1) How have you selected the healthy control patients and obtained the endometrial samples? Usually, over 55-years-old females have passed the final menstruation, which make so hard to collect the endometrium. Furthermore, the endometrium from the menopaused female contains relatively less glandular cells compared to pre-menopausal female, which is differ from the breast or lung tissue.

2) As you mentioned in LL321-323, soluble PD-1 and PD-L2 level showed significantly differ age-dependent manner. Actually, these protein level seems associated with age in the ‘case group’, however, as you have not adjusted the age in the ‘control group’, it is impossible compared between two groups.

3) In the process of RNA extraction from the endometrial tissue of control female, have you checked the infiltration of immune cells preliminary? The infiltration of immune cells in the uterine endometrium shows variety during the menstrual cycles in the pre-menopaused female, which may distort your data.

**Reviewer #2: ** thank you for allowing me to review this paper

The paper investigate the role of PDL1 expression in ec and its role

1-There is a strong correlation between pdl1 expression, TMB, and MMR status. It means that just having MMR status is useful to select patients who fit for ICI. Please comment on this

2- In the introduction of the paper is important to comment on the recent, cumulative data on ICI in first line setting (PMID: 38431043).

3- How about the role of molecular classification

4- How about PDL1 in multiple classsfier endometrial cancer . please discuss this point

5- How about p53 wt and the potential role of selinexor (PMID: 37271639) in patients with and without Pdl1 expression

6. PLOS authors have the option to publish the peer review history of their article (what does this mean? ). If published, this will include your full peer review and any attached files.

**Do you want your identity to be public for this peer review?** For information about this choice, including consent withdrawal, please see our Privacy Policy .

Reviewer #1: No

Reviewer #2: No

---

## [Author Response · Author response to Decision Letter 0]

2 Sep 2024

Response to comments raised by Editors and Reviewers on manuscript PONE-D-24-25776 titled: Gene Expression and Soluble Protein Level of PD-1 and Its Ligands (PD-L1 and PD-L2) in Endometrial Cancer

Dear Editor

We appreciate the editors and reviewers for taking the time to review our manuscript PONE-D-24-25776 and the valuable comments. Below are our responses to the comments in numerical order. We made the corrections according to the suggestions and attached the main manuscript, both with track changes and clean copy.

Review Comments to the Author and Response

Reviewer #1:

Present article is so interesting and must be valuable for the patients suffering from uterine endometrial cancer. However, several concerns remain in this article, which suggest as follows:

Comment 1:

How have you selected the healthy control patients and obtained the endometrial samples? Usually, over 55-years-old females have passed the final menstruation, which make so hard to collect the endometrium. Furthermore, the endometrium from the menopaused female contains relatively less glandular cells compared to pre-menopausal female, which is differ from the breast or lung tissue.

Response 1:

Thank you for the comment. Details have been added as needed. For the control group, samples were collected from benign cases with no significant uterine or ovarian pathology detected on imaging. Endometrial sampling, performed using a pipelle, will be histologically confirmed as normal endometrium. This sentence has been added to the manuscript, lines 122-134. Regarding the age range of the control group, it was between 30 and 60 years.

Comment 2:

As you mentioned in LL321-323, soluble PD-1 and PD-L2 level showed significantly differ age-dependent manner. Actually, these protein level seems associated with age in the ‘case group’, however, as you have not adjusted the age in the ‘control group’, it is impossible compared between two groups.

Response 2:

All samples in the control group were under 60 years old, which precludes it from being considered a subgroup since the cancer group includes ages both below and above 60. The manuscript has been updated to include information about this limitation, specifically addressing the challenge of obtaining samples from individuals older than 60 for the control group (see lines 465-466).

Comment 3:

In the process of RNA extraction from the endometrial tissue of control female, have you checked the infiltration of immune cells preliminary? The infiltration of immune cells in the uterine endometrium shows variety during the menstrual cycles in the pre-menopaused female, which may distort your data.

Response 3:

Control samples collected are from women aged between 30 and 60 years old, with half of them being menopausal (see line 198-199). Before RNA extraction, the pathologist evaluated the tissue samples by performing H&E staining on a tissue section to inspect for the presence and distribution of immune cells visually (info added in text as necessary, in line 141-144).

Reviewer #2:

Thank you for allowing me to review this paper. The paper investigate the role of PDL1 expression in ec and its role

Comment 1:

There is a strong correlation between pdl1 expression, TMB, and MMR status. It means that just having MMR status is useful to select patients who fit for ICI. Please comment on this.

Response 1:

The necessary information has been added in discussion, as indicated in lines 444-450.

Comment 2:

In the introduction of the paper is important to comment on the recent, cumulative data on ICI in first line setting (PMID: 38431043).

Response 2:

Thank you for the suggestion. The necessary information has been added in introduction, as indicated in lines 91-104.

Comment 3:

How about the role of molecular classification

Response 3:

Thank you for the suggestion. The necessary information has been added in discussion as line 426-436.

Comment 4:

How about PDL1 in multiple classsfier endometrial cancer. please discuss this point

Response 4:

Thank you for the suggestion. The necessary information has been added in discussion, as indicated in line 437-443.

Comment 5:

How about p53 wt and the potential role of selinexor (PMID: 37271639) in patients with and without Pdl1 expression

Response 5:

Thank you for the suggestion. The necessary information has been added in line 451-461.

Thank you.

Best Regards,

Dr. Chew Kah Teik

---

## [Decision Letter · Decision Letter 1]

14 Oct 2024

Gene Expression and Soluble Protein Level of PD-1 and Its Ligands (PD-L1 and PD-L2) in Endometrial Cancer

PONE-D-24-25776R1

Dear Dr. Chew,

We’re pleased to inform you that your manuscript has been judged scientifically suitable for publication and will be formally accepted for publication once it meets all outstanding technical requirements.

Kind regards,

Yasuhiro Miki

Academic Editor

PLOS ONE

Additional Editor Comments (optional):

Reviewers' comments:

Reviewer's Responses to Questions

**Comments to the Author**

1. If the authors have adequately addressed your comments raised in a previous round of review and you feel that this manuscript is now acceptable for publication, you may indicate that here to bypass the “Comments to the Author” section, enter your conflict of interest statement in the “Confidential to Editor” section, and submit your "Accept" recommendation.

Reviewer #1: All comments have been addressed

2. Is the manuscript technically sound, and do the data support the conclusions?

Reviewer #1: Yes

3. Has the statistical analysis been performed appropriately and rigorously? 

Reviewer #1: Yes

4. Have the authors made all data underlying the findings in their manuscript fully available?

Reviewer #1: Yes

5. Is the manuscript presented in an intelligible fashion and written in standard English?

Reviewer #1: Yes

6. Review Comments to the Author

Reviewer #1: Thank you for correcting your manuscript. I am fully satisfied with all your responses regarding the inquiries.

7. PLOS authors have the option to publish the peer review history of their article (what does this mean? ). If published, this will include your full peer review and any attached files.

**Do you want your identity to be public for this peer review?** For information about this choice, including consent withdrawal, please see our Privacy Policy .

Reviewer #1: No

---

## [Editor Report · Acceptance letter]

PONE-D-24-25776R1

PLOS ONE

Dear Dr. Chew,

I'm pleased to inform you that your manuscript has been deemed suitable for publication in PLOS ONE. Congratulations! Your manuscript is now being handed over to our production team.

Kind regards,

on behalf of

Dr. Yasuhiro Miki

Academic Editor

PLOS ONE